# Accelerating assimilation development for new observing systems using EFSO

Guo-Yuan Lien[1,2], Daisuke Hotta[3,1], Eugenia Kalnay[1], Takemasa Miyoshi[2,1,4], Tse-Chun Chen[1]

[1]Department of Atmospheric and Oceanic Science, University of Maryland, College Park, Maryland, 20742, USA
[2]RIKEN Advanced Institute for Computational Science, Kobe, 650-0047, Japan
[3]Meteorological Research Institute, Japan Meteorological Agency, Tsukuba, 305-0052, Japan
[4]Japan Agency for Marine-Earth Science and Technology (JAMSTEC), Yokohama, 236-0001, Japan

*Correspondence to*: Guo-Yuan Lien (guo-yuan.lien@riken.jp)

**Abstract.** To successfully assimilate data from a new observing system, it is necessary to develop appropriate data selection strategies, assimilating only the generally useful data. This development work is usually done by trial-and-error using Observing System Experiments (OSEs), which are very time- and resource-consuming. This study proposes a new, efficient methodology to accelerate the development using the Ensemble Forecast Sensitivity to Observations (EFSO). First, non-cycled assimilation of the new observation data is conducted to compute EFSO diagnostics for each observation within a large sample. Second, the average EFSO conditionally sampled in terms of various factors is computed. Third, potential data selection criteria are designed based on the non-cycled EFSO statistics, and tested in cycled OSEs to verify the actual assimilation impact. The usefulness of this method is demonstrated with the assimilation of satellite precipitation data. It is shown that the EFSO based method can efficiently suggest data selection criteria that significantly improve the assimilation results.

## 1 Introduction

Improvements in Numerical Weather Prediction (NWP) depend fundamentally on the efficient assimilation of available observations. Technological advances in remote sensing have introduced a growing number of new observing systems. However, in most cases, assimilation of a new observing system is a difficult task: naively assimilating every observation usually degrades the forecasts. It is necessary to implement an appropriate data selection process, such as selection based on channels, locations, data quality flags, and background conditions, to assimilate mostly useful data that improve model forecasts. Sometimes this data selection process is called, or overlapped with the process of quality control (QC). However, the "intrinsic" quality of the observational data is usually not the only reason for their "usefulness" in data assimilation (DA). Therefore, in this article, we use a more general term "data selection criteria" to refer to all data selection processes prior to the assimilation of a dataset.

A common approach to test the impact of assimilating a new set of observations is to perform Observing System Experiments (OSEs), which compare two otherwise identical experiments, one which includes the assimilation of the new

observing system (denoted TEST) and the other which does not (denoted CONTROL). This approach has a serious shortcoming: the signal from new observations in TEST is obscured by the presence of the many observations that are already assimilated in CONTROL, making it difficult to discern the impact from the newly assimilated observations. As a result, in order to detect statistically significant signals, it is usually required to conduct experiments over a long period of time (Geer, 2016). This makes any trial-and-error tuning of the data selection criteria by OSEs computationally very expensive and slow. Methods to accelerate this assimilation development processes are thus needed.

Forecast Sensitivity to Observations (FSO) is a diagnostic technique that allows estimating how much each individual observation improved or degraded the forecast. It was introduced by Langland and Baker (2004) for a variational DA system using an adjoint formulation. This technique was then adopted in many operational and research NWP systems (e.g., Cardinali, 2009; Gelaro and Zhu, 2009; Ishibashi, 2010; Lorenc and Marriott, 2014; Zhang et al., 2015) and has turned out to be a powerful diagnostic tool. Its ensemble formulation, Ensemble Forecast Sensitivity to Observations (EFSO), was introduced by Liu and Kalnay (2008) and Li et al. (2010) for the Local Ensemble Transform Kalman Filter (LETKF; Hunt et al., 2007). It was then refined by Kalnay et al. (2012) so that it is applicable to any formulation of ensemble Kalman Filter (EnKF), and it is simpler, easier to implement, and more accurate than the previous ensemble formulations. Ota et al. (2013) successfully implemented this new formulation to a quasi-operational global EnKF system of National Centers for Environmental Prediction (NCEP), and Sommer and Weissmann (2014, 2016) applied the method to a regional convective-scale LETKF system. Compared to the adjoint based FSO, one great advantage of the ensemble based EFSO method is that it does not require the tangent linear model, which makes the implementation easier, especially for moist processes that are very difficult to linearize. However, the approximation of nonlinear error propagation using the ensemble error covariance with a limited ensemble size may introduce some errors. To suppress the spurious correlations, spatial localization of the covariance is needed in EFSO. The covariance localization further leads to the potential necessity of advection of the localization function, although this is technically easily solvable (Kalnay et al., 2012; Ota et al., 2013).

Since the (E)FSO method can estimate the assimilation impact of any each observation at the same time, it is much more economical than conducting many OSEs, so that the idea of using (E)FSO to aid the assimilation development is attractive. However, several characteristics of FSO may restrict the effectiveness of this approach. For example, Todling (2013) pointed out that the typically used 24-hour forecast error reduction measured by a linearized total energy norm in FSO may not suitably reflect the general impact on forecasts through the 6-h-cycle DA. Besides, the use of analyses made by the own system as the verifying truth and the validity of the linear assumption in the adjoint model (or ensemble error covariance) can further make the (E)FSO estimates inaccurate. Gelaro and Zhu (2009) also discussed several differences between the FSO and OSEs. They compared the observation impacts estimated by FSO with those obtained from actual data-denial OSEs and concluded that they were only in reasonable agreement if the relative impact of a subset of observation to the total impact was considered. Therefore, the (E)FSO methods have been used mostly to monitor the performance of operational systems rather than to aid the assimilation development.

However, recent studies have demonstrated that it is indeed possible to use the EFSO information to detect and reject detrimental observations and improve the forecasts. Ota et al. (2013) showed that by applying EFSO to relatively small horizontal regions, observations that cause significant forecast degradation can be identified. Following their achievement, Hotta et al. (2017a) proposed a novel QC algorithm, termed "Proactive QC", which detects detrimental observations after only 6 hours from the analysis using EFSO and then repeats the analysis and forecast without using the identified data. They also showed that EFSO is applicable not only to a pure EnKF but also to hybrid variational-ensemble DA systems and that the EFSO results are rather insensitive to the choice of verifying truth and evaluation lead-time. The proactive QC is a method to apply EFSO *online* in DA cycles, which can adapt to the latest changes of the observing systems. However, since it requires future observation to compute the EFSO, it is applicable only to improve the "final analysis" but not to improve the "early analysis" that provides initial conditions to the extended forecasts.

In this study, we also explore the use of EFSO for observation selection (or QC), but in an *offline* approach. In contrast to the proactive QC, the EFSO is computed by offline assimilation of the data from a new observing system over a long period, and then the statistics of these EFSO samples are used to efficiently determine optimal data selection criteria for the new observing system. Section 2 describes the EFSO algorithm. Section 3 outlines our proposed methodology using EFSO to avoid expensive trial-and-error OSE experiments in the assimilation development. In Section 4, the method is demonstrated with the assimilation of a global precipitation dataset, known as TRMM Multisatellite Precipitation Analysis (TMPA; Huffman et al., 2007, 2010). Section 5 summarizes this study and concludes with a discussion.

## 2 EFSO formulation

This section briefly reviews the EFSO formulation following Kalnay et al. (2012). Let us assume that our DA system has an assimilation window of 6 hours and that we wish to quantify contribution from each of the observations assimilated at time 0 to the reduction (or increase) of the error of the forecast $t$ hours later. Let $\overline{\mathbf{x}}_{t|-6}$ and $\overline{\mathbf{x}}_{t|0}$ denote the ensemble mean forecasts valid at the evaluation time $t$ that are initialized, respectively, at time $-6$ and 0 (that is, before and after the assimilation at time 0), $\mathbf{x}_t^v$ denote the verifying state at time $t$, and $\mathbf{C}$ is a positive definite matrix that defines the forecast error norm. The change of the forecast errors due to the assimilation is measured by

$$\Delta e^2 = \mathbf{e}_{t|0}^{\mathrm{T}}\mathbf{C}\mathbf{e}_{t|0} - \mathbf{e}_{t|-6}^{\mathrm{T}}\mathbf{C}\mathbf{e}_{t|-6} \ , \tag{1}$$

where $\quad \mathbf{e}_{t|0} = \overline{\mathbf{x}}_{t|0} - \mathbf{x}_t^v, \quad \mathbf{e}_{t|-6} = \overline{\mathbf{x}}_{t|-6} - \mathbf{x}_t^v \tag{2}$

are the forecast errors after and before the assimilation, respectively. In the EnKF formulation, Kalnay et al. (2012) showed that Eq. (1) can be approximated as:

$$\Delta e^2 \approx \delta \mathbf{y}^{\mathrm{T}} \frac{\partial\left(\Delta e^2\right)}{\partial \mathbf{y}} \tag{3}$$

,

where
$$\frac{\partial\left(\Delta e^2\right)}{\partial \mathbf{y}} = \frac{1}{K-1} \mathbf{R}^{-1} \mathbf{Y}^a \mathbf{X}_{t|0}^{f\,\mathrm{T}} \mathbf{C}\left(\mathbf{e}_{t|0} + \mathbf{e}_{t|-6}\right) \tag{4}$$
.

[cf., Eq. (6) in Kalnay et al. (2012)] Here $K$ is the ensemble size, $\mathbf{R}$ is the observation error covariance matrix. $\mathbf{Y}^a = \mathbf{H}\mathbf{X}^a$ is the matrix of analysis perturbations in observation space where $\mathbf{H}$ is the Jacobian of the observation operator $H$, $\mathbf{X}^a$ is the matrix made up with $K$ column vectors representing the analysis perturbation at time 0. In practice, the $\mathbf{Y}^a$ can be computed either by applying the observation operator to the analysis members, or by applying the EnKF

analysis equation to $\mathbf{Y}^b = H(\mathbf{X}_{0|-6}^b)$ when the former method is not applicable. $\mathbf{X}_{t|0}^f$ is like $\mathbf{X}^a$ but with each column vector representing the ensemble forecast perturbation valid at time $t$ initialized at time 0. $\delta \mathbf{y}$ is the observation-minus-background (O-B) innovation vector at time 0 defined by $\delta \mathbf{y} = \mathbf{y}^o - \overline{H(\mathbf{x}_{0|-6})}$, where $\mathbf{y}^o$ is the column vector composed of the observations assimilated at time 0.

In practice, as with any EnKF with an ensemble size small compared with the rank of the covariance, covariance

localization is necessary. We need to localize the (cross-)covariance $\frac{1}{K-1} \mathbf{Y}^a \mathbf{X}_{t|0}^{f\,\mathrm{T}}$ so that Eq. (4) becomes

$$\frac{\partial\left(\Delta e^2\right)}{\partial \mathbf{y}} = \frac{1}{K-1} \mathbf{R}^{-1} \left[\rho \circ \left(\mathbf{Y}^a \mathbf{X}_{t|0}^{f\,\mathrm{T}}\right)\right] \mathbf{C}\left(\mathbf{e}_{t|0} + \mathbf{e}_{t|-6}\right) \tag{5}$$

,

Where the symbol $\circ$ represents element-wise multiplication (Schur product) and $\rho$ is a matrix whose $j$-th row is a localization function around the $j$-th observation.

Equation (3) can be interpreted as an inner product of the innovation vector $\delta \mathbf{y}$ and the sensitivity vector $\frac{\partial\left(\Delta e^2\right)}{\partial \mathbf{y}}$, so

that the contribution from a single observation, say the $l$-th element of the observation vector $\mathbf{y}^o$, can be expressed as

$$\left(\Delta e^2\right)\big|_{\left(\mathbf{y}^o\right)_l} \approx \left(\delta \mathbf{y}\right)_l \left[\frac{\partial\left(\Delta e^2\right)}{\partial \mathbf{y}}\right]_l \tag{6}$$
.

This is the EFSO estimate of the impact of a single observation $\left(\mathbf{y}^o\right)_l$ onto the $t$-hour forecast.

The EFSO can be computed based on different error norms. In this study, we use both a dry total energy norm and a moist total energy norm (Ehrendorfer et al., 1999). The generic moist total energy norm is defined as

$$\mathbf{e}^{\mathrm{T}}\mathbf{Ce} = \frac{1}{2}\frac{1}{S}\int_S\left[\int_0^1\left(u'^2 + v'^2 + \frac{C_p}{T_r}T'^2 + \frac{L^2}{C_pT_r}q'^2\right)d\sigma + \frac{R_dT_r}{P_r^2}P_s'^2\right]dS \tag{7}$$

where $S$ represents the target region, $\sigma$ is the vertical sigma-coordinate and $u'$, $v'$, $T'$, $q'$ and $P_s'$ denote, respectively, zonal and meridional wind, temperature, specific humidity and surface pressure of the perturbation $\mathbf{e}$. $C_p$, $R_d$

and $L$ are the specific heat of the air at constant pressure, the gas constant of the dry air, and the latent heat of condensation per unit mass, respectively. $T_r$ and $P_r$ are the reference temperature and surface pressure, for which we use constant values of 280 K and 1000 hPa, respectively. The dry total energy norm is defined as in Eq. (7) but excluding the moisture term [ $\left(L^2/C_pT_r\right)q'^2$ ].

## 3 Methodology to use EFSO for developing data selection criteria

The objective is to accelerate the process to determine observation selection criteria with which data from a new observing system can be effectively assimilated through an offline EFSO calculation. To achieve this, some important characteristics of the (E)FSO need to be first noted.

It has been shown that relative values of the (E)FSO among different sets of observations can be reasonably accurate compared to the true forecast impact (Gelaro and Zhu, 2009; Ota et al., 2013), despite some limitations (Gelaro and Zhu,

2009; Todling, 2013; Kalnay et al., 2012). However, it is important to note that the EFSO calculation assumes a given background state (first guess) $\mathbf{x}_{0|-6}$, a given set of observations $\mathbf{y}^o$ assimilated at the analysis time, and a given verifying state $\mathbf{x}_t^v$ at the evaluation time. If any of them changes, the EFSO for individual observations also changes. Therefore, we should not consider EFSO some kind of "intrinsic properties" or "quality" of the observation itself. The EFSO depends on the background state and the other observations assimilated at the same time. Adding a new type of observation may

decrease the impact of the others. Particularly, an important consideration to our objective is that, if cycled DA experiments are performed with different sets of observations, the changes in the background state can accumulate and thus one becomes different from another cycled DA experiment. In these circumstances, the EFSO value of a single observation is subject to stochastic fluctuations. To extract useful information, computing statistics over a sufficient number of observations becomes necessary.

In the method of proactive QC (Hotta et al., 2017a), since the EFSO is computed following each (online) DA cycle, the above conditions are mostly met. Specifically, the background state does not change when the data that passed the EFSO-

based QC are assimilated again, which makes the EFSO for individual observations useful. However, when the EFSO is computed offline (i.e., independent of the cycled OSEs), EFSO statistics over many observations need to be made. Besides, to make the EFSO as close to the true forecast impact as possible, the changes of such conditions should be minimized. With these caveats in mind, we propose the procedure for using EFSO in the assimilation development as follows:

**Step 1.** Computing EFSO samples for the new observing system using an offline DA.

    The flowchart in this step is shown in Fig. 1. Firstly, the CONTROL experiment is conducted, running DA cycles without using the new observing system, just as in the OSE framework. The ensemble first guesses (blue lines in Fig. 1) have to be saved at this time. Next, initiated from every one or several cycles in CONTROL, an "offline" (i.e. non-cycled) DA is
10   performed (purple dotted lines in Fig. 1), assimilating all data from the new observing system (with no or minimal data screening) in addition to those assimilated in the CONTROL. Using a 6-hour evaluation time, EFSO for each single observation from the new observing system are then computed using the formulation described in Section 2, which involves additional 6-hour ensemble forecasts from the non-cycled analyses (red lines in Fig. 1) and an ensemble mean forecast from the first guess (cyan lines in Fig. 1). Hotta et al. (2017a) showed that a short 6-hour evaluation time for EFSO is sufficient,
15   which is good for saving computational cost and minimizing the nonlinear effect. Note that screening (QC) the new observations should be avoided in this offline assimilation because we want to obtain EFSO for all new observations, whether they improve the forecast or not. The offline assimilation in each cycle is independent: Once the EFSO computation is done, the offline analyses are dropped (not cycled). A large sample of the single-observation EFSO data is thus collected from these multiple offline assimilation cycles.

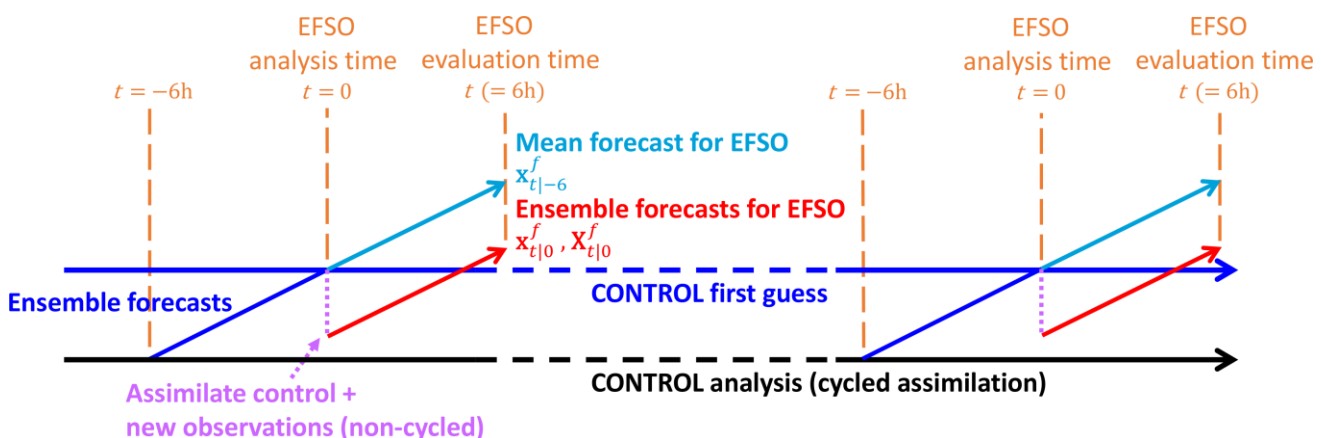

**Figure 1: Flowchart of the "offline" non-cycled EFSO computation.**

**Step 2.** Investigating the EFSO statistics for possible data selection strategies.

Having the samples of EFSO for the new observations, we assess their average impacts, such as the average EFSO for a single observation (hereafter "per-obs EFSO") or the rate of observations having positive impact (negative change in forecast errors), conditionally sampled based on various factors that could affect the assimilation impact of the observations. These factors should be simple so that they can be potentially used to formulate regular data selection criteria in the assimilation process. Examples of these factors can include: (1) geographical or vertical locations, (2) local time or phase of the diurnal/seasonal cycles, (3) channels for satellite radiance observations, (4) any kind of data quality flags provided with the data, (5) observed values or O-B departures, (6) in an ensemble DA system, statistical properties from the background ensemble, such as the number of precipitating members if assimilating precipitation, and (7) other meteorological conditions such as presence or absence of precipitation or clouds. Taking the average impacts from large samples reduces the stochastic fluctuations. These results tell us the relative usefulness in the assimilation among the new observations in terms of the chosen factors. Note that, as discussed before, the EFSO, which can be computed with various error norms, is not precisely identical to the actual forecast impact in the NWP, so it should not be expected that a set of precise optimal data selection criteria can be readily determined from the EFSO results. However, based on the comprehensive EFSO information with respect to many different factors, we can propose some data selection strategies that can be used in the assimilation of the new data, by keeping observations leading to larger beneficial average EFSO impacts and rejecting observations leading to smaller or even detrimental average EFSO impacts.

**Step 3.** Verifying the actual forecast impact by OSEs.

The data selection criteria proposed from Step 2 using the EFSO statistics need to be finally tested in regular OSEs, where subsets of new observations passing the criteria are assimilated in addition to those assimilated in CONTROL with DA cycles. The development is thought to be successful if the forecasts in TEST are better than that in CONTROL. If the forecasts are not improved, fine tuning of the selection criteria or other possible selection strategies suggested from Step 2 can be tried. This step is similar to running trial-and-error TEST experiments described in Section 1 without any EFSO information; however, with the valuable guidance from the EFSO statistics, the numbers of trials needed to obtain the optimal data selection criteria should be considerably reduced.

We note that the above design is intended to increase the representativeness of the offline computed EFSO. The simultaneous assimilation of all observations already used in CONTROL minimizes the change in the observation set $\mathbf{y}^{o}$, and the non-cycled design minimizes the change in the background state $\mathbf{x}_{0|-6}$ due to potentially many detrimental observations in the new observing system. At this initial development stage, it is assumed that we do not know a good way to effectively assimilate the new data, so it is better to avoid accumulating their possibly detrimental effects to background states with cycled DA. In addition, it would be desirable, if possible, to use another analysis dataset that is independent to

CONTROL as the verifying truth to compute the EFSO, so that the errors caused by the undesirable correlations between the forecasts and the verifying analyses are avoided (Todling, 2013).

We further note that if the OSE results are not satisfactory, an iteration of the entire three-step procedure can be considered. The procedure is repeated by letting the new CONTROL assimilate the observations used in the previous CONTROL plus the new observations that pass the first-order data selection criteria derived in the previous time. This brings the new background state $\mathbf{X}_{0|-6}$ in the EFSO calculation closer to that in actual OSEs, allowing for more accurate estimation of the observation impacts, which should benefit the determination of the final optimal data selection criteria.

## 4 A demonstration with TMPA precipitation assimilation

In order to test the methodology proposed above, we use a global satellite precipitation dataset, TMPA (Huffman et al., 2007, 2010), as an example of a new observing system, and demonstrate how this method can work to efficiently formulate appropriate data selection criteria for the new data.

### 4.1 Background on the precipitation assimilation studies

Satellite precipitation estimates, such as TMPA, have not been widely used in DA because of several problems, including the non-Gaussian error distribution associated with precipitation, the error covariance between precipitation and other model variables, and the substantial model and observation errors (e.g., Errico et al., 2007; Tsuyuki and Miyoshi, 2007; Bauer et al., 2011; Lien et al., 2016a). Assimilation of all precipitation data without special treatments usually leads to no impacts or negative impacts. In spite of these issues, Lien et al. (2013, 2016b) and Kotsuki et al. (2017) conducted a series of experiments, from an idealized configuration to realistic systems, to show that it is possible to improve the medium-range forecasts in a global model by the assimilation of global precipitation data. The keys in their experiments are to use

(1) An LETKF to exploit the flow-dependent background error correlation between prognostic variables and diagnosed precipitation,

(2) A Gaussian transformation of the precipitation variable, which mitigates the inherent non-Gaussianity of precipitation data, and

(3) Proper data selection criteria to exclude the "bad" observations that we cannot effectively use, including the important requirement that enough ensemble background members should have non-zero precipitation (Lien et al., 2013).

The great difficulty of assimilating precipitation data makes it an ideal example to demonstrate our proposed method to accelerate the assimilation development for a new observing system.

## 4.2 Experimental design

We use the same system as in Lien et al. (2016b), assimilating the global TMPA data into a low-resolution version of the NCEP Global Forecast System (GFS) model with the LETKF. The model resolution is spectral T62 with 64 vertical levels. Thirty-two ensemble members and the 6-hour assimilation cycle are used. The CONTROL experiment assimilates the rawinsonde observations processed in the NCEP Global Data Assimilation System (i.e., contained in the NCEP PREPBUFR dataset). It is conducted for 13 months from 0000 UTC 1 December 2007 to 0000 UTC 1 January 2009. It is identical to the "RAOBS" experiment in Lien et al. (2016b), where more details can be found. This CONTROL serves as the basis of the following non-cycled EFSO calculation (Step 1), and as the reference of the cycled OSEs (Step 3). We note that in an operational system already assimilating more data, the improvement that additional assimilation of precipitation can achieve should be smaller.

For the TMPA precipitation assimilation, we also employ all the techniques developed in Lien et al. (2016a, 2016b). Namely, the TMPA data are upscaled to the low-resolution T62 GFS grids and temporally integrated into 6-hour accumulation amounts. The Gaussian transformations for the model/observation precipitation variables [i.e., the "GTbz" method in Lien et al. (2016b)] are applied. Other details of the experimental configuration and the Gaussian transformation method are discussed in Lien et al. (2016b). These techniques are essential to achieve a basic satisfactory performance of the TMPA assimilation; however, even with them, assimilating all available data does not lead to an optimal result. Lien et al. (2013, 2016b) determined the data selection criteria that can further improve the assimilation results by many trial-and-error OSEs. Here we assume no knowledge of the selection criteria, and we derive them using the proposed offline EFSO methodology.

## 4.3 Step 1: Computing EFSO samples for the new observing system using an offline DA

As described above, the CONTROL experiment is cycled assimilation of rawinsonde observations. We then perform non-cycled offline assimilation of both the rawinsonde and TMPA to compute EFSO samples of the TMPA data, following the schematic in Fig. 1. Almost all available TMPA data are assimilated except those few whose innovation [difference between the observation and the model background value, $\left| \mathbf{y}^o - \overline{H(\mathbf{x}^b)} \right|$ ] is greater than 5 times of the observation errors. The European Centre for Medium-range Weather Forecasts (ECMWF) ERA Interim reanalysis is used as an independent verifying truth for EFSO computation. We follow the suggestion in Hotta et al. (2017a) to evaluate the forecast sensitivity at 6-hour forecast time, while the localization function is not advected in this study. We assume that the 6-hour forecast window is short enough so that the impact of localization advection is not essential. EFSO measured by both the moist total energy and dry total energy norms are computed. We note that for this purpose, EFSO has an advantage with respect to FSO, because FSO requires an adjoint model which has difficulties with moist processes (Janiskova and Cardinali, 2016); in addition, the use of the Gaussian transformation for assimilating precipitation in our experiments should also cause

difficulties if using an adjoint method. We perform the offline assimilation and collect EFSO samples once every 5 cycles (30 hours) over the entire year of 2008. Only every 5 cycles are computed for EFSO diagnostics in order to save computational cost while avoiding sampling the same hour in all the diurnal cycles. In total, we collect EFSO diagnostics for every single precipitation observation in 293 offline cycles, which amount to about $2.9 \times 10^6$ samples. The EFSO is computed

using naturally the same number of ensemble members, 32, as in the DA experiments. It may be speculated whether such a small ensemble size is enough to produce useful EFSO results; thus, we also compute the EFSO with even fewer ensemble members to examine if the results significantly change, which will be discussed later.

### 4.4 Step 2: Investigating the EFSO statistics for possible data selection strategies

The EFSO statistics can be conditionally computed based on various factors that can be potentially used to formulate data

selection criteria. Based on the characteristics of the precipitation data, Lien et al. (2013) suggested that when there are too many non-precipitating background members, the ensemble background error covariance may not contain enough useful information for effective DA. Other factors that may affect the effectiveness of assimilation include the observed precipitation values, particularly whether they are zero or not, and the geographic locations of the observations. Therefore, we calculate the EFSO statistics based on these factors.

Figure 2 shows the per-obs EFSO (first row) and the rate of observations with positive impacts (second row), grouped by the number of precipitating members in the background. The temporally averaged total EFSO (i.e., total EFSO divided by the number of offline cycles, 293) in one cycle is also shown (bars in the third row), as well as the total observation numbers (in all 293 times) in each group (red lines in the third row). First, it is found that the EFSO measured by the moist total energy norm shows both larger per-obs forecast error reductions and higher positive impact rates than that measured by the

dry total energy norm. This is an encouraging result, but it is not unexpected because the precipitation observation should contain useful information related to moisture. However, the assimilation of precipitation can also improve the other dynamical variables if there are at least several (greater than about 10) precipitating background members, as seen in the EFSO values with the dry total energy norm. An explanation of these results is that when fewer members are precipitating, the assimilation is more difficult because there are fewer ensemble members with dynamics closer to the truth, and therefore

the analysis increment is more uncertain. For both norms, the per-obs EFSO reaches its maximum when the number of precipitating members is around 22 (Fig. 2a,b). When the precipitating members are too many, the per-obs EFSO slightly reduces, possibly because the background states are already on average more accurate in this situation. Measured with the moist (dry) total energy norm (Fig. 2c,d), the percentages of beneficial observations increase gradually from about 52% (50%) when the number of precipitating members is less than 16 to about 54% (53%) when it is more than 20, and the

overall beneficial rate for all precipitation observations is 53.5% (51.8%), which is consistent to the general experience that this rate is usually slightly above 50% (e.g., Gelaro et al., 2010). When accumulating all observations in one cycle, the total beneficial impact shown by the total EFSO (i.e., per-obs EFSO times observation numbers; bars in Fig. 2e,f) increases roughly monotonically with the number of precipitating members, due to the effect of observation numbers: except for the

case of no precipitation in all members, there are more observations when there are many precipitating members (red lines in Fig. 2e,f).

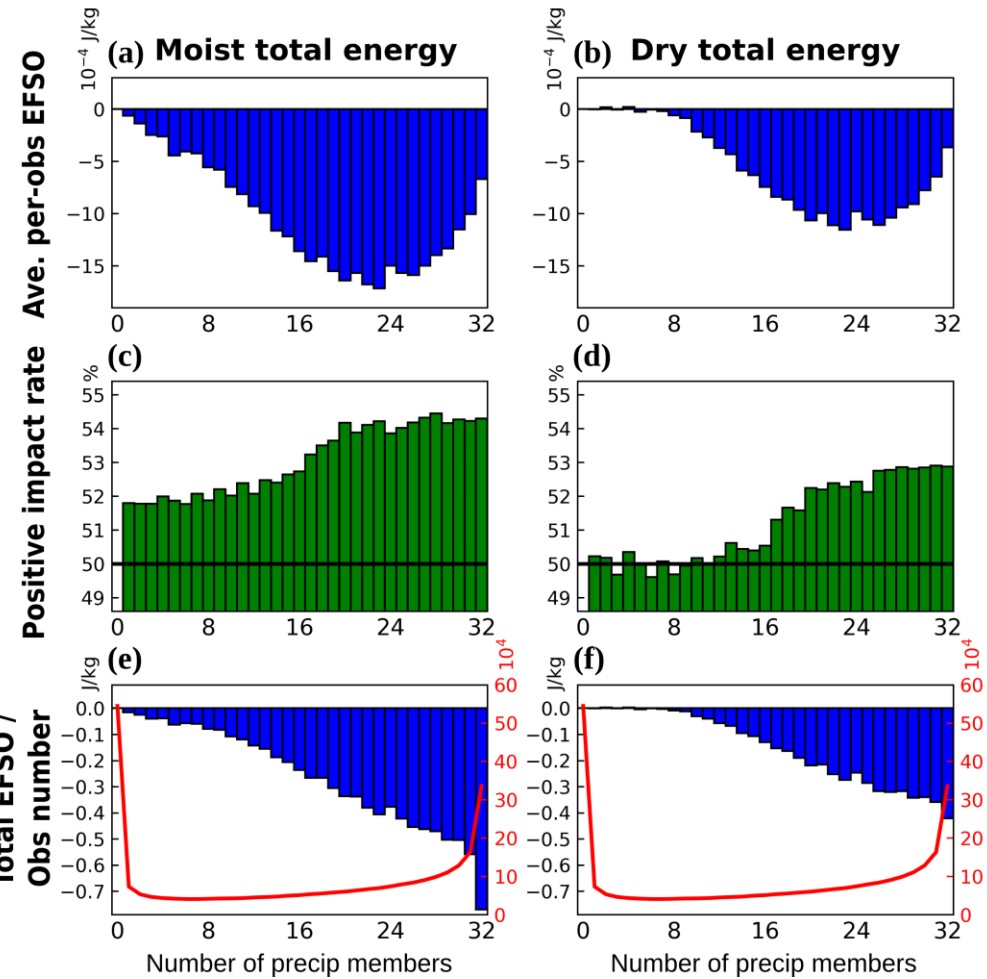

Figure 2: EFSO statistics for TMPA observations grouped by the number of precipitating members in the background and measured by (first column) the moist total energy norm, and (second column) the dry total energy norm in 6-hour forecasts during the year 2008. (a)–(b): average per-obs EFSO ($10^{-4}$ J kg$^{-1}$). (c)–(d): percentage of observations with positive impacts. (e)–(f): total EFSO per cycle (J kg$^{-1}$). Also shown in red curves in (e)–(f) are the total numbers of observations (i.e., EFSO samples) in all 293 offline cycles ($10^4$; secondary y-axis).

In addition, the EFSO statistics using the moist total energy norm are also computed separately under the condition of nonzero precipitation in the observation (hereafter R>0; first column in Fig. 3) and zero precipitation in the observation (hereafter R=0; second column in Fig. 3). For R>0 observations, the average per-obs EFSO (Fig. 3a) is all beneficial (reduction of the error), but it is most beneficial when about half of the ensemble forecasts are precipitating and half are not, indicating that the observations of precipitation are most useful when there is high uncertainty in the forecasts. The

percentage of the R>0 observations that are beneficial (Fig. 3c) is astonishingly high, reaching almost 70% under the condition that the number of precipitating members is between 5 and 13. Similarly it shows lower percentages with fewer or more precipitating members, but still much above 50%. We note that in a realistic data assimilation system such very high beneficial rates should only be found when taking subsets of observations as in this example. Besides, the relatively

5 inaccurate rawinsonde-only CONTROL, which allows the precipitation observations to contribute a larger amount of information, would be another reason for this high beneficial rate. With a modern operational system, such extremely high rates would be more difficult to be seen. The situation is quite different for R=0 observations: If the number of precipitating ensemble members is 20 or less, assimilating the R=0 observations has a detrimental effect, and it only becomes beneficial when most of the ensemble members are (wrongly) precipitating (Fig. 3b). The percentage of beneficial R=0 observations is

10 less than 50% unless essentially all the ensemble members are precipitating (Fig. 3d).

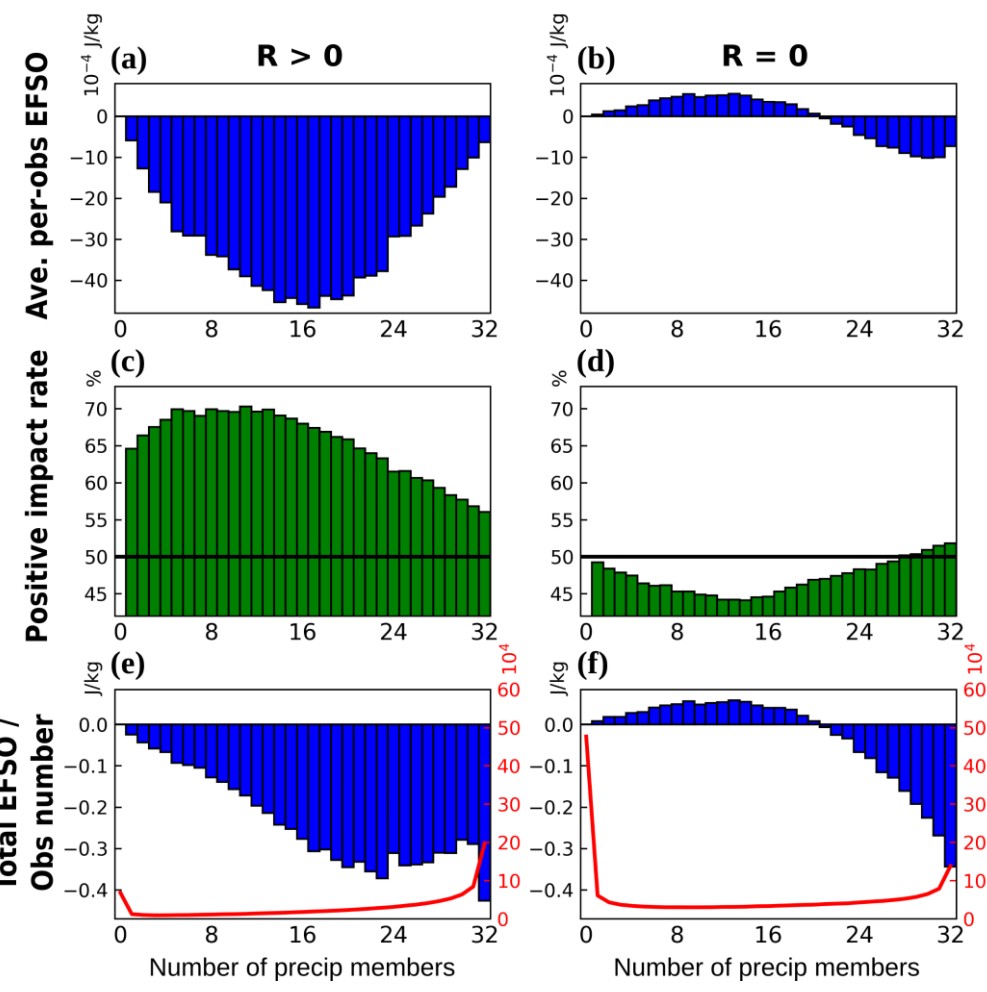

**Figure 3: Same as Fig. 2, but for EFSO measured by the moist total energy norm and computed separately for (first column) non-zero precipitation observations (R>0) and (second column) zero precipitation observations (R=0).**

To validate if these EFSO statistics computed based on the 32-member ensemble are robust, we compute the same EFSO statistics using only the first 8, 16, and 24 members. We note that this check only varies the number of ensemble members for the EFSO computation but not that in the (offline) DA; namely, the precipitation observations are still assimilated using 32 members, but the EFSO is computed [Eq. (5)] using ensemble forecasts of fewer members to the evaluation time ($\mathbf{X}_{t|0}^{f}$).

Figures like Fig. 3 but computed from fewer members are shown in Supplement. Comparing Fig. 3 and Figs. S1–3 in Supplement, we conclude that the average per-obs EFSO statistics over the large samples hardly change even with a very small ensemble size, 8 members, but the rates of beneficial observations become generally closer to 50% with fewer members. We think that the insensitivity of per-obs EFSO to the ensemble sizes is due to the average over large samples from multiple cycles, that overcomes the errors in individual observations. In contrast, the beneficial rates are more sensitive to the ensemble size because small errors in near-neutral impact observations can easily change their signs. However, for the purpose of this work, since the important information we like to know from these EFSO statistics is just the qualitative usefulness among different groups of observations, an ensemble size of 32 or even fewer is shown to be enough for the EFSO computation given the sufficient sample size.

Next, Fig. 4a,b shows the EFSO statistics (using the moist total energy norm) with respect to the geographic locations. Overall, the areas benefitted the most by the precipitation assimilation are the storm-track regions, located within 30–50°N and °S over the three major oceans. Most of the ocean regions show positive impacts and greater-than-50% positive impact rates. The marine stratocumulus regions are an exception showing detrimental impacts over the ocean. The land regions show marginal or negative impacts. These two different measures show generally similar patterns, but the detrimental regions over the land are more clearly highlighted with the positive impact rate. Here we show an interesting comparison of these EFSO maps to the correlation map between the 6-hour accumulated precipitation in the 3 to 9-hour T62 GFS model forecasts and the TMPA observations at the corresponding times (Fig. 4c), which was obtained in Lien et al. (2016a). Note that Fig. 4c is similar to Fig. 10 in Lien et al. (2016a) but for all seasons combined. This correlation score represents a simple measure of the statistical "consistency" between the model and the observation climatology, whereas details on this correlation calculation are described in Lien et al. (2016a). It was hypothesized in Lien et al. (2016a) that the precipitation observations distributed over the regions with higher correlations could be more useful for data assimilation, but that the data over very low correlation regions would be difficult to be used mainly because of large model errors. Here a great similarity is found between the average EFSO map (Fig. 4a) and the correlation map (Fig. 4c), indicating that the hypothesis in Lien et al. (2016a) was reasonable, and that the effectiveness of the precipitation assimilation is in fact strongly dependent on the geographic locations, which can be explained by the systematic inconsistency between model and observed precipitation (Lien et al., 2016a). We note again that the intrinsic quality of each observation is not the only reason for its good or bad EFSO or assimilation impact.

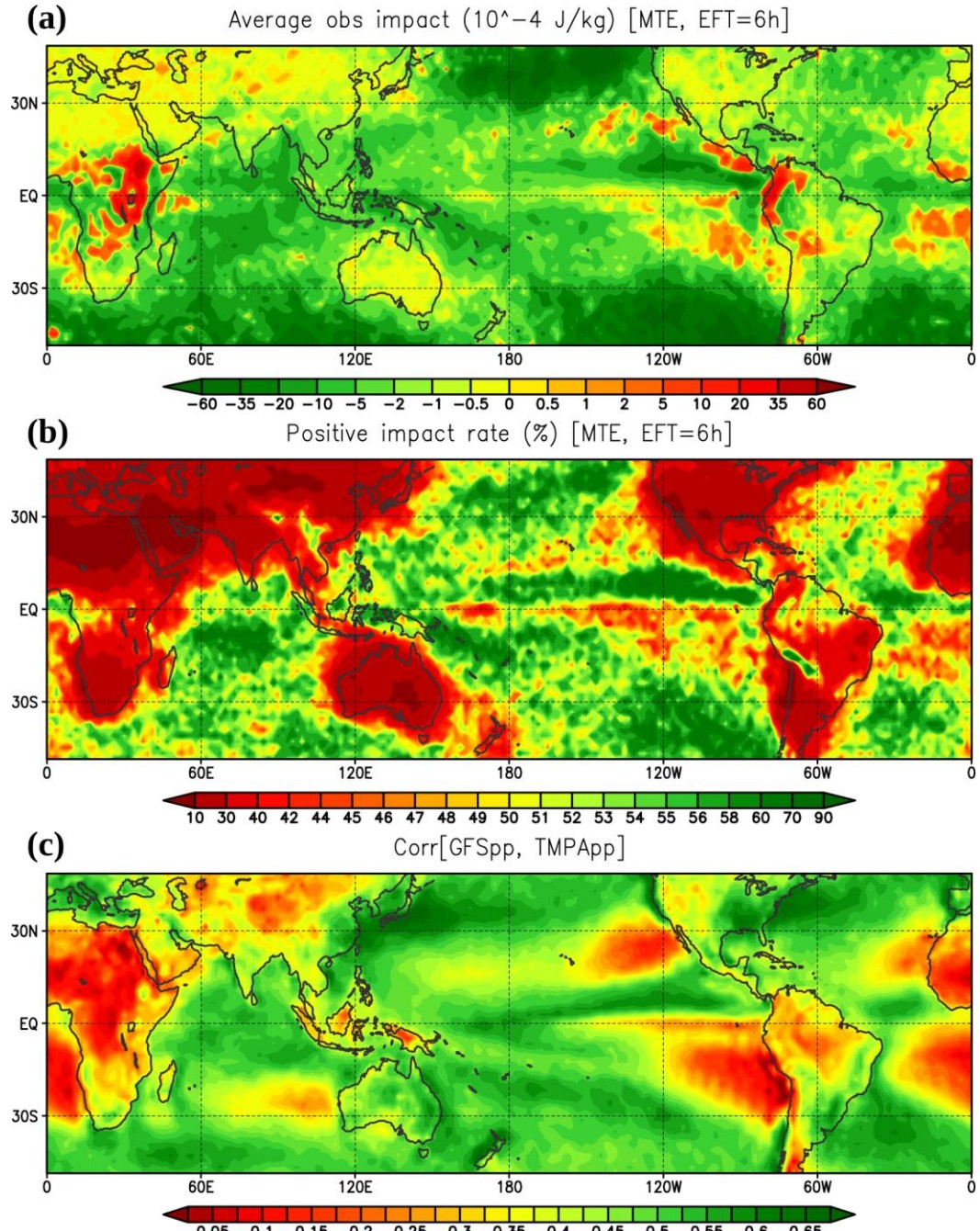

**Figure 4: The maps of (a) the average per-obs EFSO ($10^{-4}$ J kg$^{-1}$) and (b) the rate (%) of observations having positive impacts from the same precipitation EFSO sample (measured by the moist total energy norm) used in Fig. 2. (c) The maps of correlation between the 6-hour accumulated precipitation in the 3 to 9-hour T62 GFS model forecasts and the TMPA observations at the corresponding times during the year 2001–2010 period. This correlation map is similar to Fig. 10 in Lien et al. (2016a) but for all seasons combined. Details on this correlation calculation are described in Lien et al. (2016a).**

## 4.5 Step 3: Verifying the actual forecast impacts by OSEs

With the guidance from the EFSO statistics obtained in Step 2, we propose several data selection criteria and verify their actual forecast impact by OSEs. We focus on the two factors on which the EFSO largely depends: the number of precipitating members in the background and whether the observed precipitation values are zero or not. First, only the impact of the number of precipitating members is considered. We know from the EFSO statistics (Figs. 2 and 3) that the precipitation assimilation may be effective only when there are enough precipitating members in the model background. Therefore, a series of OSEs using different threshold of the number of background precipitating members for data selection are performed:

- **ALL**: all the precipitation observations are assimilated irrespective of the number of precipitating members.
- **8mR**: the precipitation observations are assimilated if at least 8 members are precipitating.
- **16mR**: As 8mR but with at least 16 members precipitating.
- **24mR**: As 8mR but with at least 24 members precipitating.

The experimental settings are summarized in Table 1. The additional 1mR/24mR experiment guided by the EFSO statistics and its results will be described later. All these experiments are conducted for the same 13-month period as in CONTROL, and 5-day forecasts are done every cycle and verified against the ERA Interim reanalysis. Considering the adjustment time required for the changing observation network, the results in the first month are discarded so that the one-year (2008) results are verified.

**Table 1: Settings of CONTROL and all TEST (i.e., ALL, 8mR, 16mR, 24mR, 1mR/24mR) experiments.**

| Experiment | | Observation assimilated | | Data selection criteria: minimum number of precipitating background members | |
|---|---|---|---|---|---|
| | | Rawinsonde | TMPA | for non-zero precip. obs. | for zero precip. obs. |
| CONTROL | | X | | N/A | N/A |
| TEST | ALL | X | X | 0 | 0 |
| | 8mR | X | X | 8 | 8 |
| | 16mR | X | X | 16 | 16 |
| | 24mR | X | X | 24 | 24 |
| | 1mR/24mR | X | X | 1 | 24 |

Figures 5 and 6 show that the assimilation of precipitation gives significant positive impacts in the global average for all these experiments, reducing the errors compared to CONTROL. In terms of global root-mean-square errors (RMSE) in 24-

hour forecasts (Fig. 5), it is found that assimilating all the observations under the requirement that at least a subgroup of members be precipitating improves the forecasts further, especially in the 500hPa zonal wind and temperature. The 24mR seems to be the optimal threshold, while assimilating too many observations (i.e., 8mR and ALL) reduces the improvement. We note that although the globally average per-obs EFSO estimates suggest mostly beneficial or neutral impacts when at

5  least one member is precipitating (Fig. 2) especially with the moist total energy norm, using a stricter criterion (i.e., 24mR) is actually more advantageous as verified in these cycled OSEs. The 24mR criterion used in Lien et al. (2016b) was obtained by the trial-and-error approach; here we re-investigate this data selection criterion from the EFSO perspective.

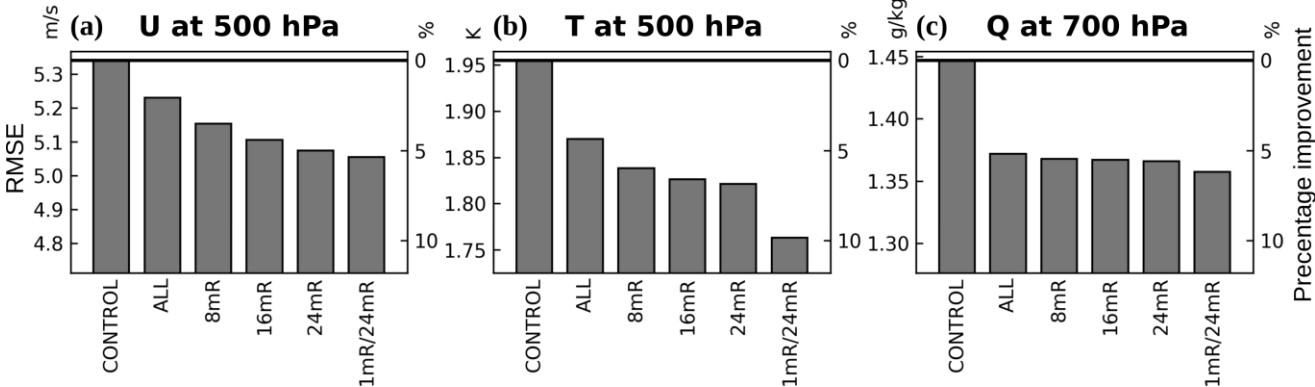

10  **Figure 5: One-year global RMSE (verified against the ERA interim reanalysis) of (a) 500-hPa u-wind (m s$^{-1}$), (b) 500-hPa temperature (K), and (c) 700-hPa specific humidity (g kg$^{-1}$) in the 24-hour forecasts for the cycled OSEs. The left axes show the absolute values; the right axes show the relative improvement to CONTROL in percentage. The thick horizontal lines indicate the RMSE in CONTROL as a reference.**

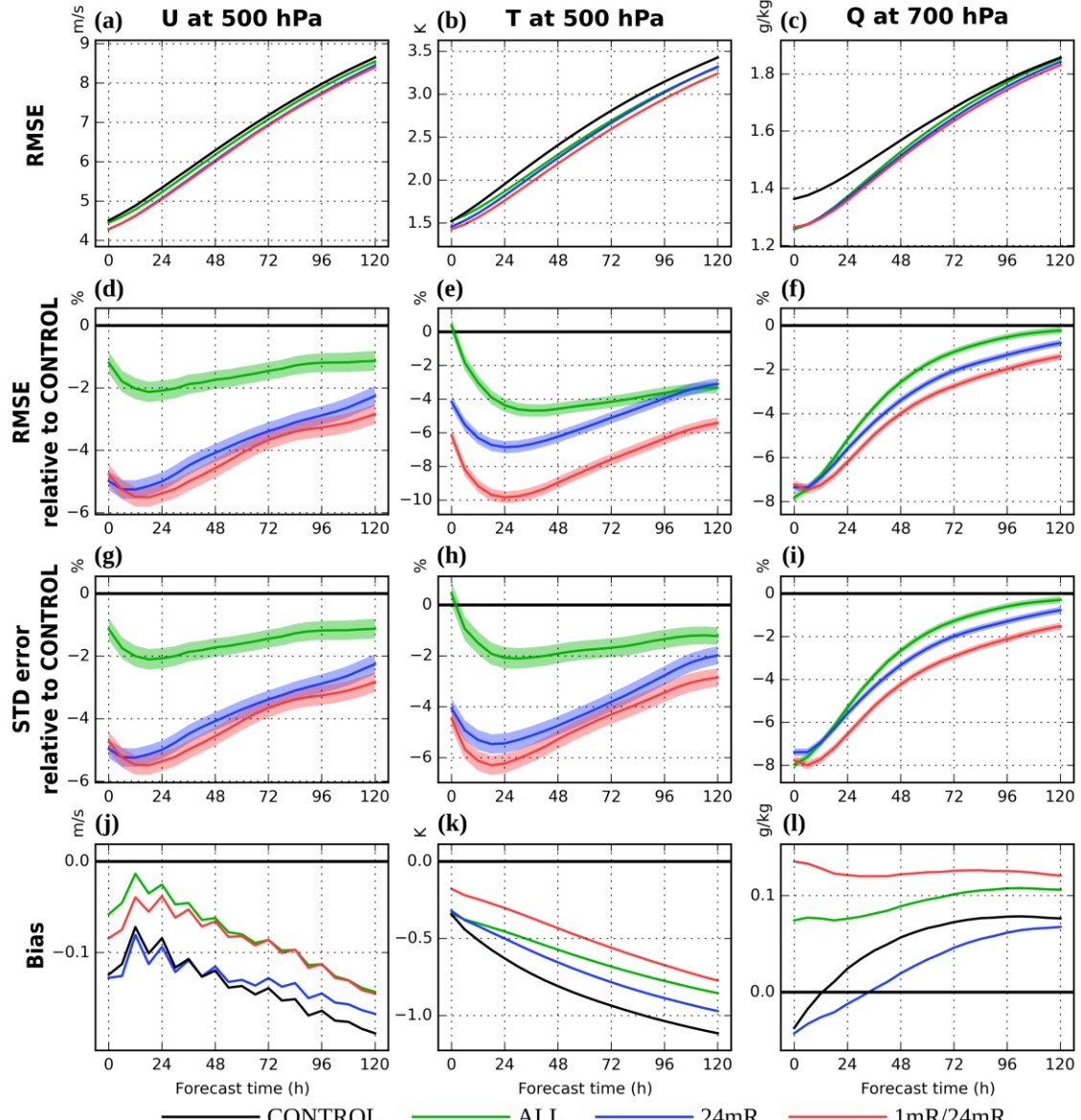

**Figure 6:** (a)–(c) RMSE, (d)–(f) RMSE relative to CONTROL (%), (g)–(i) standard deviations of errors (i.e., random errors) relative to CONTROL (%), and (j)–(l) biases during the 5-day forecasts of (first column) 500-hPa u-wind, (second column) 500-hPa temperature, and (third column) 700-hPa specific humidity for the cycled OSEs. They are verified globally against the ERA interim reanalysis over one-year period. The light color shades associated with each experiment indicates the 95% significance interval compared to CONTROL based on a pair-difference t-test.

Figure 6 shows the 0- to 5-day forecast RMSEs and biases versus the forecast time for a subset of experiments. Compared to CONTROL, the ALL (green) and 24mR (blue) experiments improve the forecasts over the entire 5-day period in all variables except for 500-hPa temperature at the analysis time in ALL, and the improvement in 24mR compared to ALL

is also clearly seen. The improvement in wind and temperature is large throughout the 5-day forecasts, while the benefit of not using the precipitation observations with fewer precipitating background members (i.e., 24mR) is significant in these dynamical variables (Fig. 6d,e). For the moisture variable, the difference between ALL and 24mR is much smaller in early forecasts, indicating that the 24mR criterion is less important for the moisture variable (Fig. 6f). This is consistent to what we found by investigating the EFSO statistics with different energy norms: to achieve effective forecast error reduction measured by the dry total energy norm (that is associated only with the dynamical variables) requires more background precipitating members than that measured by the moist total energy norm. Note that these results are statistically significant as seen by the lighter color shades associated with each experiment indicating the 95% significance interval compared to CONTROL based on a pair-difference *t*-test (e.g., Geer, 2016).

We mentioned that the "24mR" criterion based on the background members using a single threshold for all observations was proposed and studied in Lien et al. (2013, 2016b) without employing this offline EFSO guidance, but the results of the current study clearly show that the EFSO statistics can indeed help our understanding of why this criterion is so useful. Furthermore, here we will demonstrate the power of the offline EFSO approach by introducing another experiment with a more detailed data selection criteria which could only be derived with the information from the EFSO statistics. Since Fig. 3 indicates that R>0 observations have particularly larger beneficial impacts compared to the other cases, we assimilate all R>0 observations as long as at least one member is precipitating (1mR). For R=0 observations, these EFSO result suggests that they should not be assimilated unless most of the ensemble members are incorrectly predicting precipitation (Fig. 3b,d), so we assimilate them only if at least 24 members are precipitating (24mR). This experiment is therefore named (Table 1):

- **1mR/24mR**.

The OSE result (5-day global forecast impact) of this two-threshold data selection criteria can also be seen in Figs. 5 and 6. This 1mR/24mR experiment based on the offline EFSO guidance outperforms any single-threshold experiment and also all the results obtained in Lien et al. (2016b) without employing the EFSO guidance. The improvement is particularly large in the 500-hPa temperature forecast, showing an additional ~3% reduction in the RMSE compared to 24mR in the 24-hour forecasts (Fig. 5b). Furthermore, although the EFSO only estimate the 6-hour forecast error reduction, the forecast improvement by this data selection lasts throughout the 5 days (Fig. 6).

To investigate further the impact of the 1mR/24mR experiments, we show the biases compared to the ERA interim reanalysis in Fig. 6j–l. The magnitudes and changes of biases by the precipitation assimilation are rather small compared to the RMSE for 500-hPa u-wind and 700-hPa moisture. However, the magnitude of 500-hPa temperature bias is large in our model compared to its RMSE, and the precipitation assimilation is correcting it in the longer forecast time. This can be additionally examined by looking at the standard deviation of errors which represents "debiased RMSEs" or "random errors", shown in Fig. 6g–i. In this bias-free verification, the 500-hPa temperature improvement in all the precipitation assimilation experiments over CONTROL becomes smaller than that measured by the RMSE, and the improvement of 1mR/24mR over 24mR also becomes smaller, because a portion of this improvement in the RMSE is achieved by reducing the bias. For 500-hPa u-wind and 700-hPa moisture, their errors are much less affected by removing their biases, meaning that their biases are

negligibly small compared to the RMSEs. It is also noted that the magnitude of the improvement in 500-hPa temperature becomes similar to that in 500-hPa u-wind, which can be suggested by geostrophic balance. Nonetheless, the 1mR/24mR is still better than 24mR in all variables, indicating the usefulness of our EFSO methodology, despite of the existence of the model bias.

The regional verifications are shown in Fig. 7. The improvement by 1mR/24mR is seen in most of the verification regions and variables, within the entire 5-day forecast period, so we believe that the optimality of 1mR/24mR is robust. Note that most of these differences in forecast RMSEs are statistically significant (shown by the light color shades). However, for tropical regions, the 1mR/24mR does not analyse well the 500-hPa u-wind and 700-hPa moisture (at the analysis time; Fig. 7g,i), but it becomes better than the other experiments with forecast length, possibly due to the improvement in other

variables/regions at the analysis time. The improvements in the tropical forecasts made with 1mR/24mR after day 2 are remarkable.

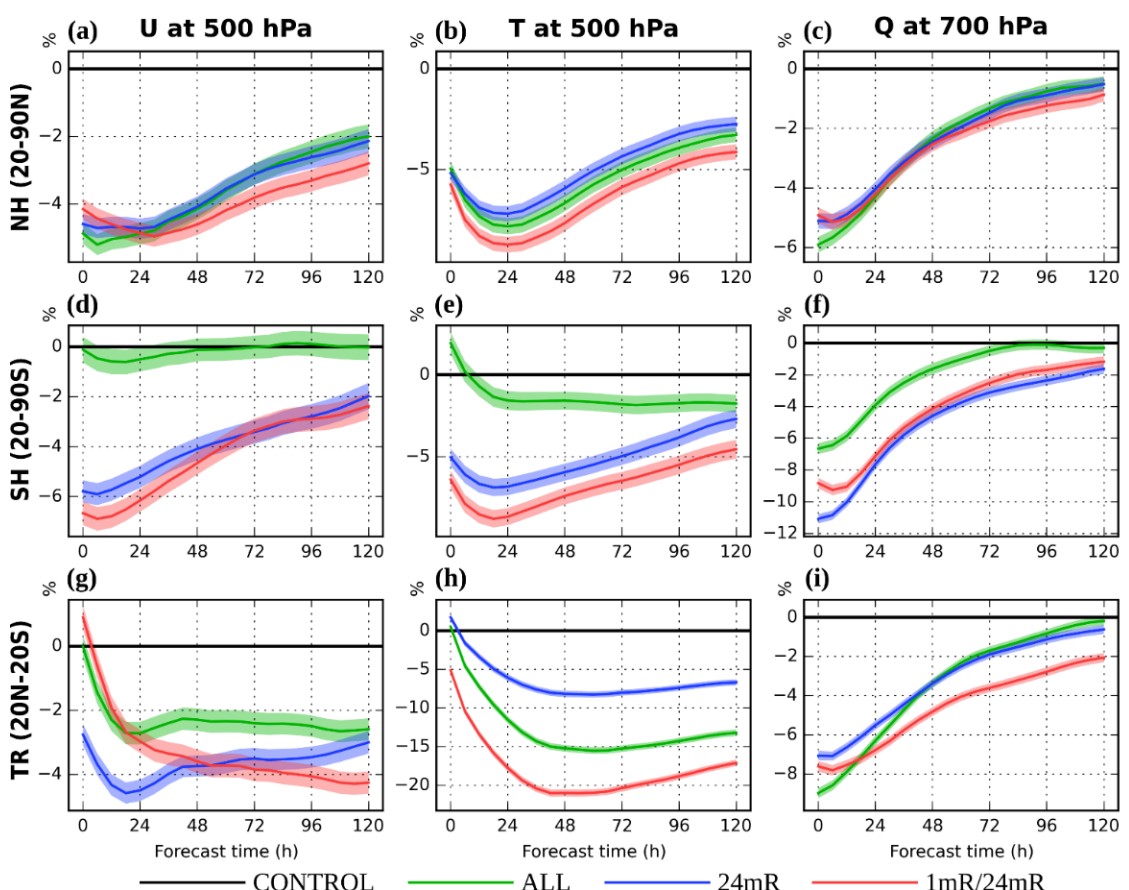

Figure 7: Same as Fig. 6, but for RMSE relative to CONTROL (%) in different verification regions: (a)–(c) Northern Hemisphere
extratropics (NH; 20–90°N), (d)–(f) Sorthern Hemisphere extratropics (SH; 20–90°S), and (g)–(i) tropics (TR; 20°N–20°S).

These experiments demonstrate the power of the EFSO approach to guide the development of the data selection strategies in assimilating new observing systems. We only need to compute the EFSO from the offline assimilation experiments once, over a sufficiently long period to collect a large sample, and then the statistics can be done comprehensively based on many different factors. These statistics can provide much useful information. For example, it would be unlikely that without knowing the EFSO statistics, one could formulate the 1mR/24mR criteria with just a few trial-and-error experiments.

Lastly, we note that we also tried to make use of the geographical dependence (Fig. 4) of EFSO to formulate the data selection criteria used in OSEs, such as 1) applying constant geographic masks defined by single thresholds of average per-obs EFSO values in Fig. 4, used in addition to the current 24mR or 1mR/24mR criteria, or 2) further separating EFSO statistics in Figs. 2 and 3 to different latitudinal bands and defining different data selection criteria for them. However, although these experiments are similarly good, none of them could outperform 1mR/24mR, which remains the best results we have obtained.

## 5 Summary and discussion

A common approach to develop the data selection strategies and criteria for assimilating a new observing system is to perform OSEs, which is usually very time- and resource-consuming. Here, we propose a new methodology based on the EFSO method, that allows us to estimate the forecast impact of each observation. We propose that the EFSO statistics from a large sample can be used to guide the development of the data selection criteria in a more straightforward manner and thus accelerate the development work.

The proposed method consists of 3 steps: 1) Conduct an "offline" (not cycled) DA experiment, assimilating most of available new observational data with no or minimal screening or QC. From the offline assimilation, compute large samples of per-obs EFSO data for the new observing system. 2) Compute the average per-obs EFSO and/or the percentage of beneficial observations conditionally sampled based on various factors that can be potentially used to formulate criteria for data selection, such as location, time, satellite channels, data quality flags, and model background conditions. Based on these statistics indicating the relative forecast impacts by the assimilation of the new observations under different conditions, potential data selection criteria can be proposed for keeping more beneficial observations. 3) Verify the actual forecast impact of the data selection criteria by applying them in cycled OSEs.

We demonstrate this method with the assimilation of TMPA global satellite precipitation data, which have been studied by Lien et al. (2016a, 2016b). It is shown that the EFSO approach supports the data selection criterion proposed in Lien et al. (2013, 2016b) based on the number of precipitating members in the model background. In addition, it further provides useful information for refining the data selection criteria, so that in this study we obtain a result significantly better than the best obtained by Lien et al. (2016b). This example shows how the EFSO method can be used to accelerate the development of an optimal data selection strategy for assimilating new observing systems.

The setup of this demonstrative experiment is just intermediate; compared with the modern operational configuration, the resolution is too low, and the observation data used in the CONTROL experiment are much limited, consisting of just rawinsondes. We believe that the use of this intermediate setup does not hinder the objective of this study, which is to demonstrate the methodology of using EFSO to accelerate the development of the quality control in data assimilation. However, in an operational system already assimilating more data, the improvement that additional assimilation of precipitation can achieve will be smaller. Regarding this aspect, we think that this method could indeed still be useful when abundant observations are already used in CONTROL, because the guidance provided by EFSO statistics is relevant, independent information that is not obscured by the existing observations. Nevertheless, experiments with more realistic configurations are required to convincingly prove the usefulness of this method in the assimilation development with a state-of-the-art operational NWP system.

On the other hand, since this EFSO based methodology provides an efficient way to clarify under what condition observations are helpful or not, we believe that it should be useful for instrument and algorithm developers to collaborate with DA scientists, so that the EFSO information can be used to improve the quality of the assimilated observations. Such a collaboration is taking place between scientists at the Universities of Maryland and Wisconsin, with the aim to identify the origin of occasional negative EFSO impacts of high latitude MODIS winds.

We note that this method works *offline* based on the static data selection criteria derived with the aid of EFSO statistics in advance of the cycling DA, whereas Proactive QC (Ota et al., 2013; Hotta et al., 2017a) works *online* based on the dynamical criteria decided by the flow-dependent EFSO diagnostics. An important issue that is not covered in this study is how to assign appropriate observation error variance for the new observation data. Using an idea similar to EFSO, Hotta et al. (2017b) presented a new ensemble sensitivity technique, called ensemble forecast sensitivity to observation error covariance (EFSR), that allows estimating how forecast errors would change by perturbing a prescribed observation error covariance matrix, thereby enabling systematic tuning of the observation error covariance for new observing systems. The EFSO-based data selection presented in this study together with EFSR can thus provide a framework for accelerating assimilation development of new observations aided by the ensemble forecast sensitivity diagnostics.

**Code availability**

The GFS-LETKF code with the EFSO computation functionality is available at https://github.com/takemasa-miyoshi/letkf .

**Data availability**

The TMPA data used in this study is TRMM 3B42, version 7, available at the NASA website, https://pmm.nasa.gov/data-access/downloads/trmm . The rawinsonde observations are extracted from the NCEP BREPBUFR observation dataset,

available at https://rda.ucar.edu/datasets/ds337.0/ . The ERA Interim reanalysis data used for verification are available at http://apps.ecmwf.int/datasets/ .

**Author contribution**

G.-Y. Lien and E. Kalnay initiated this study at University of Maryland. G.-Y. Lien developed the global precipitation assimilation system with the low-resolution NCEP GFS model and developed the idea of "EFSO from offline assimilation." He then designed and conducted all experiments after moving to RIKEN Advanced Institute for Computational Science (AICS). D. Hotta initially implemented the EFSO diagnostics into the system and shared valuable thoughts to the methodology of using EFSO for assimilation development. E. Kalnay and T. Miyoshi were the advisors of this work. E. Kalnay provided insightful suggestions to this study and improved the final manuscript. T.-C. Chen shared thoughts to the interpretation of the EFSO and the relation between this study to the proactive QC method.

**Acknowledgements**

This study was initially conducted at University of Maryland, partially supported by NASA Grants NNX11AH39G, NNX11AL25G, and NNX13AG68G; NOAA Grants NA10OAR4310248, CICS-PAEKLETKF11, and NA16NWS4680009; and the Office of Naval Research (ONR) Grant N000141010149 under the National Oceanographic Partnership Program (NOPP). After G.-Y. Lien moved to RIKEN AICS, the work continued with additional support by JST CREST Grant JPMJCR1312 and the Japan Aerospace Exploration Agency (JAXA) Precipitation Measuring Mission (PMM). The authors thank Yoichiro Ota for providing an initial EFSO code he implemented in another DA system as a reference of the EFSO implementation. We also thank two anonymous referees and the editor, Z. Toth, for their insightful comments and suggestions.

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
