# Peer review of "Accelerating assimilation development for new observing systems using EFSO"

_Nonlinear Processes in Geophysics, 2017_

## Referee Comment (RC1) · Anonymous Referee #1 · 20 Sep 2017

This research investigated an efficient methodology to accelerate the development for appropriate data selection strategies for new observing systems using the Ensemble Forecast Sensitivity to Observations (EFSO). The EFSO diagnostics are used to design potential data selection rules for data selection. The usefulness of this method is demonstrated with the assimilation of satellite precipitation data in a low resolution global model. It is shown that the EFSO based method can efficiently aid data selection that significantly improve the assimilation and forecasting results. The manuscript is well written and easy to follow. I suggest to accept it for publication after some revisions.

Comments 1. The key for EFSO is to evaluate gradient of error reduction to observations (Eq. 4 and 6). Could you please justify why an ensemble of 32 members can

provide reasonable/accurate estimation of the gradient to precipitation? Precipitation processes are highly nonlinear, which may cause difficulties when adjoint method is used to calculate the gradient as mentioned in this manuscript and many published articles. Do you think the EFSO can deal well with this nonlinear issue? 2. It is good to see precipitation assimilation can improve the forecasts up to 5 days. But overall, RMSEs in wind and temperature are reduced about 3-5 % up to 5 days. Do you think this result can be reproduced at operational centers, e.g. NCEP, if the satellite precipitation was assimilated? 3. Experimental design. Rather than a year run using a low resolution model (T62) with an ensemble of 32 members, results from a seasonal run of high resolution model and more ensemble members that are close to operational configurations might be more convinced.

Other comments: 1. Page 2, lines 15-24. Though adjoint-based FSO method faces some difficulties, for specific applications, it is still can give a good evaluation compared to OSE. Here is an example: Zhang, X.-Y., H. Wang, X.-Y. Huang, F. Gao, and N. A. Jacobs, 2015: Using adjoint-based forecast sensitivity method to evaluate TAMDAR data impacts on regional forecasts. Adv. Meteor., 2015, 427616, doi:https://doi.org/10.1155/2015/427616. 2. Page 9, lines 18-19 Though there are some difficulties, adjoint models have been used in the operational data assimilation systems at ECMWF and UK Met office, which produce world-best global weather forecasts. EnKF has its merits but also has its limitations. The key for EFSO is to evaluate gradient of error reduction to observations (Eq. 4). Do you think an ensemble of 32 members can provide a good estimation of the gradient?

3. Figure 4. Can we assume precipitation obs over open waters have same quality? If Yes, however, the percentages of useful data from Pacific tropical region are much different. The explanation is reasonable and acceptable. Just wonder it might be good to mention that suggest to mention that rejected data, does not always mean they are bad data.

4. Figure 6. It is good to see precipitation assimilation can improve the forecasts up to

5 days. But overall, RMSEs in wind and temperature are improved about 3-5 % up to 5 days. Do you think the impact is overestimated?

Interactive
comment

---

## Author Comment (AC1) · 4 Oct 2017

[NPG-2017-45]
Response to Referee #1
October 4, 2017

We gratefully thank Referee #1 for the constructive comments. The point-by-point responses to the comments are detailed below.

**This research investigated an efficient methodology to accelerate the development for appropriate data selection strategies for new observing systems using the Ensemble Forecast Sensitivity to Observations (EFSO). The EFSO diagnostics are used to design potential data selection rules for data selection. The usefulness of this method is demonstrated with the assimilation of satellite precipitation data in a low resolution global model. It is shown that the EFSO based method can efficiently aid data selection that significantly improve the assimilation and forecasting results. The manuscript is well written and easy to follow. I suggest to accept it for publication after some revisions.**

**Comments 1. The key for EFSO is to evaluate gradient of error reduction to observations (Eq. 4 and 6). Could you please justify why an ensemble of 32 members can provide reasonable/accurate estimation of the gradient to precipitation? Precipitation processes are highly nonlinear, which may cause difficulties when adjoint method is used to calculate the gradient as mentioned in this manuscript and many published articles. Do you think the EFSO can deal well with this nonlinear issue?**

We agree that the limited ensemble size is an issue for EFSO. On one hand, our experience is that although an ensemble size of 32 is relatively small, it could still be sufficient to produce reasonable results. In previous EFSO studies, Ota et al. (2013) and Hotta et al. (2017) used 80 ensemble members, Sommer and Weissmann (2014, 2016) used 32 and 40 ensemble members. In particular, the latter two studies focused on the application of EFSO in convective scales, and they obtained useful results.

On the other hand, we believe that our choice of a short evaluation forecast time, which is only 6 hours, is helpful to mitigate the nonlinearity issue. Within the 6 hours the nonlinear effect would not be too strong. Hotta et al. (2017) showed that the results using 6-hour evaluation forecast time are qualitatively similar to those using longer evaluation times, which is very beneficial to the practical use of the EFSO method.

In addition, a crucial element that leads to the success of our precipitation assimilation study is the use of Gaussian transformation. Without the Gaussian transformation the results were bad (Lien et al., 2016). Therefore, we need to compute the forecast sensitivity when the Gaussian transformation of the precipitation variable is performed. The authors speculate that the adjoint FSO would be able to produce reasonable sensitivity gradient in this case; however, the EFSO

can still deal it with, which is one advantage of the EFSO method in this nonlinear problem.

**2. It is good to see precipitation assimilation can improve the forecasts up to 5 days. But overall, RMSEs in wind and temperature are reduced about 3-5 % up to 5 days. Do you think this result can be reproduced at operational centers, e.g. NCEP, if the satellite precipitation was assimilated?**

In our experiments, the impact of precipitation assimilation can be large because we used a CONTROL experiment assimilating only the rawinsonde data. In a modern operational NWP system that already assimilated many other conventional and satellite data, we believe that the impact should be much smaller. This design of CONTROL was also used in Lien et al. (2016), and we believe that it does not affect the main points of the current study. We will add this explanation in the revised manuscript after the open-discussion period.

**3. Experimental design. Rather than a year run using a low resolution model (T62) with an ensemble of 32 members, results from a seasonal run of high resolution model and more ensemble members that are close to operational configurations might be more convinced.**

We agree that the resolution is too low and the ensemble size is relatively small compared to the operational configuration. However, we think that the use of this intermediate setup does not hinder the objective of this study to demonstrate the methodology of using EFSO to accelerate the development of the quality control in data assimilation. Regarding the topic of precipitation assimilation, Kotsuki et al. (2017) recently showed that it is also useful with a global model at 112-km resolution (~T120) and with 36 ensemble members, although this topic is not the main focus of the current study.

**Other comments: 1. Page 2, lines 15-24. Though adjoint-based FSO method faces some difficulties, for specific applications, it is still can give a good evaluation compared to OSE. Here is an example: Zhang, X.-Y., H. Wang, X.-Y. Huang, F. Gao, and N. A. Jacobs, 2015: Using adjoint-based forecast sensitivity method to evaluate TAMDAR data impacts on regional forecasts. Adv. Meteor., 2015, 427616, doi:https://doi.org/10.1155/2015/427616.**

Thank you for pointing this out. We strongly agree that both FSO and EFSO are extremely useful. The main advantage of the ensemble based EFSO method, compared to the adjoint based FSO, is that the former does not require the development of the tangent linear model and adjoint model, which makes the implementation easier, especially for moist processes that are very difficult to linearize. We will make this clear in the revised manuscript and cite the reference you kindly provided.

**2. Page 9, lines 18-19 Though there are some difficulties, adjoint models have been used in the operational data assimilation systems at ECMWF and UK Met office, which produce world-best global weather forecasts. EnKF has its merits but also has its limitations. The key for EFSO is to evaluate gradient of error reduction to observations (Eq. 4). Do you think an ensemble of 32 members can provide a good estimation of the gradient?**

We do not deny the usefulness of the adjoint FSO. Regarding the issue of limited ensemble size, please see the response for Comment 1 above.

**3. Figure 4. Can we assume precipitation obs over open waters have same quality? If Yes, however, the percentages of useful data from Pacific tropical region are much different. The explanation is reasonable and acceptable. Just wonder it might be good to mention that suggest to mention that rejected data, does not always mean they are bad data.**

Thank you very much for pointing out the important point that "the data rejected (for improving the model forecasts) are not always bad-quality data." We believe that we have attempted to describe this point in the submitted manuscript:

"Therefore, we should not consider EFSO some kind of "intrinsic properties" or "quality" of the observation itself. The EFSO depends on the background state and the other observations assimilated at the same time." (P.5, L.9-11)

We will elaborate this point more clearly in the revised manuscript.

**4. Figure 6. It is good to see precipitation assimilation can improve the forecasts up to 5 days. But overall, RMSEs in wind and temperature are improved about 3-5 % up to 5 days. Do you think the impact is overestimated?**

Same as the response for Comment 2 above.

**References**

Hotta, D., Chen, T.-C., Kalnay, E., Ota, Y. and Miyoshi, T.: Proactive QC: a fully flow-dependent quality control scheme based on EFSO, Mon. Wea. Rev., doi:10.1175/MWR-D-16-0290.1, 2017.

Kotsuki, S., Miyoshi, T., Terasaki, K., Lien, G.-Y. and Kalnay, E.: Assimilating the global satellite mapping of precipitation data with the Nonhydrostatic Icosahedral Atmospheric Model (NICAM), J. Geophys. Res. Atmos., 122(2), 2016JD025355, doi:10.1002/2016JD025355, 2017.

Lien, G.-Y., Miyoshi, T. and Kalnay, E.: Assimilation of TRMM Multisatellite Precipitation Analysis with a low-resolution NCEP Global Forecast System, Mon. Wea. Rev., 144(2), 643–661, doi:10.1175/MWR-D-15-0149.1, 2016.

Ota, Y., Derber, J. C., Kalnay, E. and Miyoshi, T.: Ensemble-based observation impact estimates using the NCEP GFS, Tellus, 65, 20038, doi:10.3402/tellusa.v65i0.20038, 2013.

Sommer, M. and Weissmann, M.: Observation impact in a convective-scale localized ensemble transform Kalman filter, Q.J.R. Meteorol. Soc., 140(685), 2672–2679, doi:10.1002/qj.2343, 2014.

Sommer, M. and Weissmann, M.: Ensemble-based approximation of observation impact using an observation-based verification metric, Tellus A: Dynamic Meteorology and Oceanography, 68(1), 27885, doi:10.3402/tellusa.v68.27885, 2016.

---

## Referee Comment (RC2) · Anonymous Referee #2 · 6 Oct 2017

The manuscript proposes an efficient method for chosing appropriate data selection criteria for new observing systems based on EFSOI. The usefulness of this approach is demonstrated with the assimilation of precipitation observations. The findings of the paper are interesting, it's very easy to read and well-written. It should be suitable for publications after addressing the following few remarks.

There is one issue that the authors should investigate a bit more. Usually, the number of beneficial observations should be slightly above 50%. When the number is much higher, I strongly suspect that the observations are correcting or compensating some model bias. Otherwise it's unlikely to achieve numbers up to 70%. Very low numbers likely indicate the opposite effect, i.e. that there is a model bias which prevents the effective use (which is also indicated by the plot of FSOI versus precipitating members).

I think it would be very interesting to investigate this more and show more bias statistics (e.g. regional plots).

Minor remarks: 1) The WMO DAOS group recently decided that the term FSOI should be used (where "I" stands for impact) instead of FSO. I recommend following this and using EFSOI. I think there is a document on the WMO website with more details. 2) I don't think QC is the appropriate term for data selection and it's potentially misleading. Why not calling it "data selection criteria" or "observation preprocessing"? 3) Introduction: It would be good to make the discussion and literature review of strengths and weaknesses of EFSOI a bit broader and more critical. The method obviously has strengths, but also some weaknesses. E.g. there is a linearization involved, there are spurious correlations, potential bias (correction) issues and observations interact (adding a new type may decrease the impact of others).

---

## Author Comment (AC2) · 22 Nov 2017

[NPG-2017-45]
Response to Referee #2
November 22, 2017

We gratefully thank Referee #2 for the constructive comments. We have revised the paper accordingly. The point-by-point responses to the comments are detailed below.

**The manuscript proposes an efficient method for choosing appropriate data selection criteria for new observing systems based on EFSOI. The usefulness of this approach is demonstrated with the assimilation of precipitation observations. The findings of the paper are interesting, it's very easy to read and well-written. It should be suitable for publications after addressing the following few remarks.**

**There is one issue that the authors should investigate a bit more. Usually, the number of beneficial observations should be slightly above 50%. When the number is much higher, I strongly suspect that the observations are correcting or compensating some model bias. Otherwise it's unlikely to achieve numbers up to 70%. Very low numbers likely indicate the opposite effect, i.e. that there is a model bias which prevents the effective use (which is also indicated by the plot of FSOI versus precipitating members). I think it would be very interesting to investigate this more and show more bias statistics (e.g. regional plots).**

We agree with Referee #2 that the assimilation of precipitation observations may be correcting or compensating some model biases. Indeed, in the manuscript we show the 5-day forecast biases in Fig. 6j-l. Compared to their RMSE (Fig. 6a-c), the bias values are relatively small for 500-hPa u-wind and 700-hPa moisture, but the model does present considerable 500-hPa temperature bias. The precipitation assimilation generally reduces model biases for 500-hPa wind and temperature. Besides, we also show that, when measuring the forecast skill by the standard deviation of errors that does not take biases into account (Fig. 6g–i) (instead of by RMSE), although the improvement by precipitation assimilation becomes smaller, all the precipitation assimilation experiments are still better than CONTROL, and 1mR/24mR is still better than 24mR in all variables. We believe that this would be sufficient to show the superiority of 1mR/24mR and the usefulness of our EFSO methodology, despite of the existence of the model bias. The related discussion can be found in P.18, L.26–P.19, L.4. We revised the discussion regarding the model biases to better elaborate this aspect.

In addition, following Referee #2's suggestion, we show the regional bias plots (Fig. R1; c.f., Fig. 6j-l) and the regional (bias-free) standard deviation of errors (Fig. R2; c.f., Fig. 7) here. The results are more complicated in different regions, but we think the general conclusion remains as what we discussed above for the global verification: There are considerable model

bias in the temperature field and the precipitation assimilation is correcting it, but in most variables and regions, the 1mR/24mR still outperforms the other experiments in the bias-free verification. We feel that these additional figures are not very essential for the paper and would be a distraction in the manuscript if included, so we would not add these figures in the manuscript.

[Figure]

Figure R1: Similar to Fig. 6j-l, but for regional biases during the 5-day forecasts of (first column) 500-hPa u-wind, (second column) 500-hPa temperature, and (third column) 700-hPa specific humidity for the cycled OSEs, verified against the ERA interim reanalysis over one-year period. (a)–(c) Northern Hemisphere extratropics (NH; 20–90°N), (d)–(f) Southern Hemisphere extratropics (SH; 20–90°S), and (g)–(i) tropics (TR; 20°N–20°S).

[Figure]

Figure R2:  Similar to Fig. 7 and Fig. R1, but for standard deviations of errors (i.e., random errors) relative to CONTROL (%) in different verification regions.

Aside from the model bias, we think that the higher rate of beneficial observations can be understood by two other reasons. First, although the overall rate of beneficial observations is usually not too much higher than 50%, when taking some subsets of observations, due to the smaller sample sizes, it should be possible to obtain higher rates than the overall rate. For example, in our offline DA experiment, the positive impact rate for all precipitation observations are merely 53.5% and 51.8% in terms of the moist total energy norm and the dry total energy norm, respectively (added in P.10, L.22–24; c.f., Fig. 2c, d). The near 70% positive impact rate is only seen when we consider only the nonzero precipitation observations under the condition that less than half of the model background members are precipitating (Fig. 3c). Second, the positive impact rate of EFSO tends to be higher when the background is not very accurate, under which circumstance the observations are able to contribute a larger amount of information. In our experimental setting the CONTROL experiment assimilates only the rawinsonde data, and the assimilation of the precipitation effectively improves the forecast skill, so a larger positive rate compared to the experience in the modern operational system would be expectable. We included the above explanation in the manuscript (P.11, L.14–P.12 L.4).

**Minor remarks: 1) The WMO DAOS group recently decided that the term FSOI should be used (where "I" stands for impact) instead of FSO. I recommend following this and using EFSOI. I think there is a document on the WMO website with more details.**

We appreciate the comment of Referee #2, but we feel that for a procedure with a long name like "Ensemble Forecast Sensitivity to Observations", adding "Impact" at the end makes it too long, without really clarifying the meaning of the procedure.

**2) I don't think QC is the appropriate term for data selection and it's potentially misleading. Why not calling it "data selection criteria" or "observation preprocessing"?**

Thank you very much for this suggestion. We have revised throughout the manuscript to use the term "data selection criteria" instead of "QC." We also modified the explanation of the use of the terminology in the beginning of the manuscript (P.1, L.24–27).

**3) Introduction: It would be good to make the discussion and literature review of strengths and weaknesses of EFSOI a bit broader and more critical. The method obviously has strengths, but also some weaknesses. E.g. there is a linearization involved, there are spurious correlations, potential bias (correction) issues and observations interact (adding a new type may decrease the impact of others).**

We added a few sentences on the strengths and weaknesses of the EFSO method in the introduction section (P.2, L.16–21). Regarding the observation interaction issue, we added a sentence in P.5, L.17–18. The same paragraph has also broadly discussed some related issues.

---

## Author Response (AR2)

We gratefully thank the editor for his very careful review and constructive comments. We have revised the paper accordingly. A track-change version of the manuscript is attached in the end. The point-by-point responses to the editor's comments are detailed below.

In addition to the editor's comments, we made a few additional changes:
1. We re-generated Figs. 2 and 3 purely to make some texts in the figures more discernible.
2. Added the "Code availability" and "Data availability" sections in the end of the manuscript.

**Response to Editor, Zoltan Toth**

**1) As to the 1st comment of Rev. #1 about the limited (32) number of ensemble members - here I do not feel you provided a direct or particularly helpful response. I understand others in the literature used similarly small size ensembles. Perhaps you may speculate why this low number may still yield useful results? Have you checked how your method may work with fewer number of members (over a shorter time period if that helps reduce computational costs)? Do you see a saturation of impact as you approach 32 numbers? Sharing any other insight with the readers may damp related questions.**

According to the editor's suggestion, we additionally computed the EFSO statistics using only the first 8, 16, and 24 members. Note that this is only for EFSO computation, but the (offline) DA part is unchanged, still with 32 members. Figures like Fig. 3 but computed from fewer members are shown in Supplement, and the related discussion is added in P.9, L.29–32 and P.13, L.1–13. We found that the average per-obs EFSO statistics over the large samples hardly change even with a very small ensemble size, 8 members; meanwhile, although the rates of beneficial observations are more sensitive to the ensemble size, the qualitative information among different groups of observations, that is important for the purpose of this work, is not much affected. We think that the insensitivity of the per-obs EFSO statistics to the ensemble sizes is due to the average over large samples from multiple cycles, that overcomes the errors in individual observations; therefore, an ensemble size of 32 or even fewer is shown to be enough for the EFSO computation given the sufficient sample size.

**2) On your response to 3rd comment of Rev. #1, and the 3rd minor remark of Rev. #2 - Both Reviewers are asking you to qualify your results in light of the significant limitations of your study (e.g., limited number of ensemble members, low resolution of system you use, limited (one type only) set of "control" observational data). How the limitations may affect the results? You may want to reflect on these and possibly other limitations in a more concerted way, perhaps as a separate paragraph as part of your discussion, conclusions, and/or abstract, cautioning the reader against drawing too broad conclusions etc? The strengths of your method are probably well laid out already. Making the limitations of the study even clearer will only add to the value of your contribution.**

Thank the editor for this valuable suggestion. We added a separate paragraph in the conclusion section (P.21, L.1–10) to better discuss the limitations of this study regarding the low model resolution and the limited observation data used in CONTROL. For the ensemble size issue, since we have shown for Comment 1) that the small ensemble size is not a problem with the current experiment settings, we do not include it as a limitation discussed in this paragraph.

**Minor editorial and other comments:**
**p.2, l.20: "although this is technically easily solvable"**

Revised accordingly.

**p2, l.19-21: awkward sentence, can you break it into 2 sentences?**

Revised accordingly.

**p5, l.17: "Adding a new type of observation"**

Revised accordingly.

**p9, l.20-22: Too long sentence, please break it up into two sentences.**

Revised accordingly.

**p11, l.13: to lend emphasis to the point you make here, suggest this change: "reaching almost 70% under the condition that the number of precipitating members is between 5 and 13."**

Revised accordingly.

[revised manuscript text omitted]

---

## Author Response (AR3)

We gratefully thank the editor for his very careful review. We have revised the paper accordingly. In addition to the editor's comments, we made some minor changes in the abstract. The responses to the editor's comments are given below. The changes in the manuscript are also attached in the end (only for pages with changes).

**Response to Editor, Zoltan Toth**

**Thank you for your response and revisions. I may be missing something, but looking at Fig 3 and the associated discussion, I do not see a reference whether you show moist or dry energy results in these figures? Please add this information into the figure legend and text (if missing from there, too). Other than that, your manuscript is ready for publication.**

Thank you very much for pointing out this mistake we made. Except for the second column of Fig. 2, all other EFSO statistics figures show results using the moist total energy norm. We added this information into the captions of Figs. 3 (P.12) and 4 (P. 14) and the texts (P.11, L.11; P.13, L.15).

[revised manuscript text omitted]